# Identification of QTLs Controlling Resistance to Anthracnose Disease in Water Yam (*Dioscorea alata*)

**DOI:** 10.3390/genes13020347

**Published:** 2022-02-14

**Authors:** Paterne Angelot Agre, Kwabena Darkwa, Bunmi Olasanmi, Olufisayo Kolade, Pierre Mournet, Ranjana Bhattacharjee, Antonio Lopez-Montes, David De Koeyer, Patrick Adebola, Lava Kumar, Robert Asiedu, Asrat Asfaw

**Affiliations:** 1International Institute of Tropical Agriculture (IITA), Ibadan 5320, Nigeria; P.Agre@cgiar.org (P.A.A.); O.Kolade@cgiar.org (O.K.); R.Bhattacharjee@cgiar.org (R.B.); p.adebola@cgiar.org (P.A.); L.Kumar@cgiar.org (L.K.); R.Asiedu@cgiar.org (R.A.); 2Savanna Agricultural Research Institute, Tamale P.O. Box TL 52, Ghana; kwabenadarkwa@gmail.com; 3Department of Agronomy, University of Ibadan, Ibadan 200284, Nigeria; bunminadeco@yahoo.com; 4Centre de Coopération Internationale en Recherche Agronomique pour le Développement, 34398 Montpellier, France; pierre.mournet@cirad.fr; 5Amelioration Génétic et Adoption des Plants Méditerranéennes et Tropical AGAP, Universisté de Montpellier, 34398 Montpellier, France; 6International Trade Centre (ITC), Addison House International Trade Fair Center, FAGE, Accra GA145, Ghana; mijuamarel@gmail.com; 7Agriculture and Agri-Food Canada, Fredericton, NB 20280, Canada; david.dekoeyer@aagr.gc.ca

**Keywords:** *Dioscorea* spp., greater yam, genetic map, marker–trait association, linkage analysis

## Abstract

Anthracnose disease caused by a fungus *Colletotrichum gloeosporioides* is the primary cause of yield loss in water yam (*Dioscorea alata*), the widely cultivated species of yam. Resistance to yam anthracnose disease (YAD) is a prime target in breeding initiatives to develop durable-resistant cultivars for sustainable management of the disease in water yam cultivation. This study aimed at tagging quantitative trait loci (QTL) for anthracnose disease resistance in a bi-parental mapping population of *D. alata.* Parent genotypes and their recombinant progenies were genotyped using the Genotyping by Sequencing (GBS) platform and phenotyped in two crop cycles for two years. A high-density genetic linkage map was built with 3184 polymorphic Single Nucleotide Polymorphism (NSP) markers well distributed across the genome, covering 1460.94 cM total length. On average, 163 SNP markers were mapped per chromosome with 0.58 genetic distances between SNPs. Four QTL regions related to yam anthracnose disease resistance were identified on three chromosomes. The proportion of phenotypic variance explained by these QTLs ranged from 29.54 to 39.40%. The QTL regions identified showed genes that code for known plant defense responses such as GDSL-like Lipase/Acylhydrolase, Protein kinase domain, and F-box protein. The results from the present study provide valuable insight into the genetic architecture of anthracnose resistance in water yam. The candidate markers identified herewith form a relevant resource to apply marker-assisted selection as an alternative to a conventional labor-intensive screening for anthracnose resistance in water yam.

## 1. Introduction

Yam (*Dioscorea* spp.) is a multi-species monocotyledonous crop widely grown in the tropics and subtropics [1]. It is the most valuable crop in West Africa, where its cultivation began 11,000 years ago [2]. Of the over 600 yam species, water yam (*D. alata*) is an extensively cultivated species worldwide [3]. In Africa, white Guinea yam (*D. rotundata*) is the most cultivated yam species, followed by water yam [3]. Yam production in West Africa is mainly by smallholder farmers, making it a significant source of farm employment and income for this group. In addition, yam plays a vital role in traditional medicine and the socio-cultural life of the people as it is involved in many key life ceremonies [4].

Water yam possesses several valuable attributes for cultivation and consumption. These include high multiplication ratio, early vigor for weed smothering, the higher genetic potential for yield (especially under low to average soil fertility), low post-harvest losses, good processing quality, and high nutritional value, including its possession of low glycemic index [5,6]. However, anthracnose disease caused by the *Colletotrichum gloeosporioides* (Penz) is the most limiting factor affecting the productivity of water yam by devastating all parts of the yam plant at every developmental stage, including leaves, stems, tubers, and seeds in many regions of the world [7,8]. Anthracnose causes mild to acute leaf necrosis, premature leaf abscission, and shoot die-back [9]. Severe infections result in defoliation, leaving naked, black, and drying vines [10]. Yield losses from the disease of up to 90% have been reported under severe conditions on different cultivars of water yam in Africa [10,11,12]. High genetic and pathogenic variances have been reported among isolates of *C*. *gloeosporioides* from different geographical locations [7,13,14], suggesting a high probability of the geographic variation in strains, some of which could be overcome existing resistance [15].

Cultural control approaches such as the use of disease-free planting materials, adjustment of plant spacing and planting dates, burying infected plant residues in the soil immediately after harvesting, intercropping, crop rotation with non-host crops, and fallowing have been used in other plant pathosystems to reduce pathogen inoculum in the field, delay disease onset, or slow disease progress [16,17]. Nonetheless, these disease management practices have not been effective for controlling anthracnose disease in water yam or result in a substantial increase in tuber yield [8], especially in disease-endemic areas. Additionally, biological control to impede or out-compete the multiplication and spread of virulent *C. gloeosporioides* strain in yam fields has been limited [18]. Chemical control can be an effective disease management approach. Still, most yam producers are smallholder growers and may not have the prerequisite technical support and finance to afford the use of fungicides [19].

Furthermore, inappropriate use of fungicides could potentially result in the development of resistant *C. gloeosporioides* strains to systemic fungicides [20] as well as detrimental environmental effects. Therefore, the best control option is developing and deploying cultivars with durable resistance to anthracnose. Substantial progress has been made to develop anthracnose resistant water yam varieties at the International Institute of Tropical Agriculture (IITA), Nigeria, and national agricultural research systems in West Africa and elsewhere through conventional breeding using phenotypic observations [3,21]. Anthracnose-resistant cultivars of yam such as TDa1425 and TDr2040 were identified at IITA [22]. In India, laboratory and field investigations also found highly resistant *D. alata* lines [23]. However, this effort is arduous and considerably slow due to the crop’s heterozygous and vegetatively propagated nature [24]. Genomics-informed breeding techniques such as molecular marker-assisted breeding and genomic selection would accelerate efforts in introgressing anthracnose resistance into preferred genetic backgrounds [3].

Earlier investigations on anthracnose disease in water yam showed that resistance is likely dominant and quantitatively inherited [25]. Efforts have also been made to identify QTL controlling yam anthracnose disease (YAD) using low-throughput molecular markers and less dense or unsaturated genetic maps such as Amplified fragment length polymorphism (AFLP) markers [5,26] and Expressed Sequence Tag—Simple Sequence Repeats (EST-SSRs) [27,28,29]. Prospects for locating additional QTLs and applying molecular breeding methods in water yam improvement programs are up-and-coming, mainly due to advances in next-generation sequencing and the recent development of the reference genome sequence of *D. rotundata* and *D. alata.* Therefore, it is imperative to apply new molecular tools to develop additional genomic resources from different genetic backgrounds to facilitate the breeding for anthracnose resistance in water yam. The objective of this study was to develop a SNP-based genetic linkage map and identify QTL for anthracnose disease resistance in a diploid bi-parental mapping population of *D. alata*. It assessed the QTL presence, positions, the effects of QTL alleles, and the underlying putative genes in the QTL vicinities responsible for anthracnose resistance in water yam.

## 2. Materials and Methods

### 2.1. Plant Materials

An F_1_ recombinant clonal population of 204 individuals derived from a single cross of TDa0500015 × TDa9900048 was used for this study. TDa0500015 (diploid) is a female breeding line showing a tolerant reaction to yam anthracnose disease, while TDa9900048 (diploid) is a male breeding line expressing a susceptible response. The two parents and their F_1_-derived recombinant clonal progenies, along with a highly susceptible cultivar (TDa92-2), were field-phenotyped in two cropping cycles for two seasons (2017 and 2018) at IITA, Ibadan research farm in Nigeria. The field experiment was carried out using a partial replicated design of three plants per genotype in 1 × 1 m planting spacing during the main rainy seasons when anthracnose incidence and severity are high. Genotypes with high plant numbers were planted in 2 replications, and susceptible reference cultivar TDa92-2 was planted as a spreader row between blocks and around the field.

### 2.2. Phenotyping

Anthracnose disease severity was scored at two months after planting and after that, fortnightly till six months. Severity was scored by visual assessment of the relative area of plant tissue affected by anthracnose using a 1–5 severity rating scale. Where, 1 = No visible symptoms of anthracnose disease or infection spot on the leaf surface; 2 = Few anthracnose spots or symptoms on 1–25% of the plant (i.e., one or two spots of less than 1 cm diameter width, and dry tissue on the leaf surface); 3 = Anthracnose symptoms covering 26–50% of the plant (i.e., one or two spots of more than 1 cm diameter width, and dry tissue on the leaf surface, small dark and no dried spots with more than 1 cm width are present); 4 = Symptoms on >50% of the plant (i.e., coalesced spots with dry tissue and covering a significant proportion of the leaf surface, areas with less than 1 cm width coalesce to more prominent spots and yellowing of green tissue is intense around the spots areas); and 5 = Severe necrosis and death of the plant (i.e., coalesced spots with dry tissue more than 1.5 cm in diameter and covering a significant proportion of the leaf surface and yellowing of the green tissue is generalized in the leaf blade) [30]. The time series severity score was recorded on individual plants in a plot. The mean anthracnose severity for a genotype in a plot was estimated by summing severity scores >1 in a plot divided by the total number of symptomatic plants.

The area under the disease progression curve (AUDPC) was estimated from the mean disease severity scores of a genotype per plot using the trapezoidal method [31]. This method discretizes the time variable and calculates the average disease intensity between each pair of adjacent time points.
(1)$$\mathrm{AUDPC}=\overset{n}{\underset{i=1}{\sum }}\left(\frac{y_{i}+y_{i+1}}{2}\right)\left(t_{i+1}-t_{i}\right)$$
where *n* = total number of observations, *y_i_* = disease severity at the t*_i_* observation, and t = time at the t*_i_* observation.

### 2.3. Genotyping

Young fresh leaf samples were collected from the 207 genotypes (204 recombinant progenies, the two parents and a check variety) and immediately dipped in dry ice. The leaves were stored at −80 °C before lyophilization. Lyophilized leaf samples were sent to CIRAD-France for DNA extraction, library construction, and Genotyping by Sequencing (GBS). DNA extraction and Genotyping by Sequencing (GBS) were performed as described in Cormier et al. [32]. GBS libraries were constructed as described by Elshire et al. [33] using PstI-MseI restriction enzymes. Sequencing was conducted on an Illumina HiSeq 3000 system Montpellier, France (150 bp, single-end reads) at the GeT-PlaGe platform in Toulouse, France.

### 2.4. Data Analyses

#### 2.4.1. Phenotype Data

Anthracnose severity score data collected at different times during the crop’s growth period were converted to AUDPC for quantitative comparison over the years. The area under disease progress curve data was subjected to mixed model analysis using lme4 library package implemented in R [34].
(2)$$Y_{ikk}=\mu +\beta_{i}+R_{ij}+G_{k}+\left(\beta_{i}{\times G}_{k}\right)+e_{ijkm}$$
where Y*_ijk_* = phenotypic value, µ = overall phenotypic mean, β*_i_* = effect of year *i*, R*_ij_* = effect of block *j* in year *i*, G*_k_* = effect of genotype *k*, (β*_i_* × G*_k_*) = effect of interaction between year *i* and genotype *k*, and e*_ijkm_* = residual. Block within-year effects were added to the model as a random variable to remove the spatial variation within the trial field. Broad sense heritability was estimated from the model to assess the proportion of phenotypic variation in the data set due to genetic effects. Phenotypic BLUE (Best Linear Unbiased Estimator) values of un-shrunken means for QTL analysis were extracted for the years and over the years.

#### 2.4.2. SNP Calling and Quality Assessment

Raw data were first filtered using a pipeline described in Scarcelli et al. [35]. Demuladapt (https://github.com/Maillol/demultadapt, accessed on 12 March 2020) was used for demultiplexing. Cutadapt 1.2.1 [36] was used to remove the adaptors and low-quality bases read with a mean quality score <30 using a free perl script https://github.com/SouthGreenPlatform/arcad-hts/blob/master/scripts/arcad_hts_2_Filter_Fastq_On_Mean_Quality.pl, accessed on 11 March 2020. For the final SNP calling, GATK was used while mapping was performed using default options of Burrows-Wheeler Aligner (BWA) [37] using the *D. alata* reference genome v2.1 [38]. The SNP quality assessment was performed using vcftools [39] and plink [40]. SNPs with low MAF <0.05 and low depth sequencing <5 were removed. This retained 7583 SNPs out of the raw 15,936 SNPs called. For the missing point, SNP markers and genotypes with high missing information >20% were removed as well.

#### 2.4.3. Genetic Map Construction

Linkage analysis was conducted using MAPpoly package [41] in the R environment [34]. A series of filtering steps were applied using all segregating markers polymorphic in at least one of the parents to construct an integrated genetic map. Chi-square (χ2) test was conducted to calculate the marker segregation ratio and exclude markers showing significant segregation distortion from map construction. The *p*-value threshold to assess the significant marker segregation distortion was set using the filter_segregation function as implemented in MAPpoly package. To construct linkage groups, the pairwise recombination fraction and LOD matrices between markers retained after the segregation test were calculated using the function est_pairwise_rf in MAPploy package. Linkage grouping was then performed using an initial LOD value of >6 obtained from αthres function in MAPpoly. The LOD value of 6.0 that established known linkage groups was then chosen as the significance criterion for multipoint linkage testing. First, for the genetic map construction, marker loci were partitioned primarily into linkage groups (LGs). Secondly, the modified logarithm of odds (MLOD) scores between markers were calculated to further confirm the robustness of markers for each LGs. Markers with MLOD scores <6 were filtered out prior to ordering. Thirdly, the highMap strategy described by Liu et al. [42] was utilized to order the Bin markers and correct genotyping errors within and between LGs. Genetic recombination fraction (RF) was estimated for the retained SNP markers to confirm the non-switch of alleles from one LG to another using “est.rf” function implemented in R/QTL [43]. To confirm the well ordering SNP markers across LG, the recombination fraction against the LOD score was then plotted and the graph was viewed using ggplot2 R package. The final GM was then constructed using R/QTL2 [43] and viewed in LinkageMapView.

#### 2.4.4. QTL Analysis

The QTL analysis was performed with mean trait value over years and linkage map constructed from the 159 recombinant clones using the Composite Interval Mapping (CIM) method in R/QTL2 package [43]. A forward and backward simple stepwise regression was run to select background markers with a significant level of *p* < 0.05. The threshold levels to declare significant QTLs were empirically determined through 1000 permutations of the data, which maintained a chromosome-wise Type I error rate of 0.05 [44] with a fixed LOD of 4 as a minimum threshold of declaring a SNP marker linked with the YAD.

The location of a QTL was described according to its LOD peak location. The proportion of phenotypic variance accounted for by each detected QTL was estimated by a single-factor analysis of variance using the General Linear Model. The additive (Add) and dominance (Dom) effects and the proportion of phenotypic variation explained (PVE%) by each QTL were estimated using the “fitqtl” function in R/QTL. The sign of the additive effect of each QTL was used to identify the origin of the favorable alleles. A simple mixed model was implemented in lmer4 package to estimate QTL interaction/environment using the identified QTL by considering the year and the SNP marker as fixed effect, while the genotypes were considered as a random effect. Markers linked with the yam anthracnose disease were then placed in the respective chromosome, and their position was viewed using Qtl/jittermap. For the gene mining, the related putative genes associated with SNP markers were searched within the upstream and downstream locations of the QTL generic feature format (GFF3) of the reference genome of *D. alata* v2.1 [38] available on https://phytozome-next.jgi.doe.gov/, accessed 1 May 2021. Functions of the different genes associated with the identified QTL were determined using the public database Interpro, European Molecular Biology Laboratory-European Bioinformatics Institute (EMBL-EBI).

## 3. Results

### 3.1. Phenotypic Variability

Significant differences (*p* < 0.05) were observed for the reaction of the progenies to YAD in both years (Table 1). The mean squares for the year and genotype-by-year interaction effects were highly significant (*p* < 0.01). The disease pressure was higher in 2018 compared to 2017. The area under disease progression curve (AUDPC) estimates ranged from 210.0 to 397.5 with an average of 245.5 in 2017, while the range was from 233.4 to 482.1 with an average of 299.8 in 2018. None of the recombinant clones demonstrated immune or highly resistant (mean severity score of 1, equivalent to AUDPC value <105) or highly susceptible (mean severity score of 5, the equivalent of AUDPC > 525) reaction to natural field infestation by anthracnose disease. However, most of the genotypes (67–92%) expressed moderate resistance to anthracnose. Broad sense heritability was high (70.64%).

### 3.2. SNP Filtering

Total of 15,936 SNP markers were identified in the parental individuals and mapping population. Filtering for minor allele frequency (MAF < 0.05), low depth sequencing (< 5), and polymorphism between the parents TDa0500015 and TDa9900048 reduced the number of SNPs to 7583 SNPs (47.6% of the raw SNPs identified). Further filtering for 20% missing data (both for the SNP and genotype) (Appendix A) and segregation distortion with chi-square test (Appendix A) identified 3257 informative markers and 179 individuals out of 204 progenies with good coverage for linkage map construction. Pairwise recombination fractions calculated for all informative markers showed high SNP markers ordering across the different linkage groups (Appendix A).

### 3.3. Linkage Mapping

A genetic map was constructed that covered all 20 linkage groups of the water yam genome (Figure 1) with a total genetic distance of 1460.94 cM. The marker order on the linkage map showed perfect genetic recombination (Appendix A) as the recombination fraction of the mapped SNP markers on linkage groups displayed a perfect alignment with no half circles between the recombination fraction and the LOD score (Appendix A). The linkage map had an average of 163 markers per linkage group or chromosome, with the highest SNP markers mapped on linkage 5. Linkage groups 6, 5, and 2 were the longest with 109.52, 109.19, and 109.17 cM, respectively, while linkage group 19 was the shortest with 33.08 cM (Table 2). The genetic gap map intervals ranged from 2.03 and 9.07 cM on chromosomes 19 and 16, respectively (Table 2).

### 3.4. QTL Identification

The QTLs detected on three chromosomes out of the 20 are presented in Table 3 and Figure 2. Two significant QTLs were detected on chromosome 7 at position 10.60 cM (between 10.596 and 19.217 cM). This QTL (Qyad-7-1) had a LOD score of 4.51 and accounted for 33.7% of the total phenotypic variation in anthracnose severity score (Table 3, Figure 2). The second QTL located on chromosome 7 (Qyad-7-2) was at position 19.21 cM (between 10.596 and 19.218 cM) at LOD score of 5.28 and accounted for 29.54% of the total phenotypic variation in anthracnose severity score. The 3rd significant QTL, Qyad-15, which explained 30.90% of the total phenotypic variance with a LOD score of 4.43 was detected at 28.80 cM on chromosome 15. The QTL on chromosome 18 (Qyad-18) was at position 61.4 cM (between 61.345 and 61.432 cM) at LOD score of 4.65 and contributed 39.40% of the total phenotypic variance. For the four markers associated with the YAD, the favorable alleles were contributed by TDa0500015 tolerant to the YAD.

The QTL region linked to YAD resistance on chromosome 7 has known genes in plant biotic stress such as DRNTG_08663.1 (GDSL-like Lipase/Acylhydrolase), DRNTG_08664.1 (Protein kinase domain), and DRNTG_23336.1 (Appendix A). Additionally, the regions within the Qyad-15 locus were related to the N-terminal α/β domain gene DRNTG_14305.1. The vicinity of Qyad-18 showed genes that code for ANTH domain Putative clathrin assembly protein (DRNTG_18245.1) and WD domain–WD40 repeat-containing protein (DRNTG_29617.1) (Appendix A).

Interaction among the four QTLs related to YAD resistance revealed significant (*p* < 0.05) QTL by QTL interaction for Qyad-7-1 and Qyad-15, Qyad7-2 and Qyad-18. In contrast, no significant variation was observed among all other QTL combinations (Table 4). Of the four QTLs related to YAD resistance, three were stable over the years and showed non-significant QTL by environment interaction (Table 5).

### 3.5. Marker Effect

The presence of allele T for loci Qyad-7-1 and Qyad-7-2 appeared to lower the AUDPC score in the evaluated population, while the presence of the alleles C tended to increase the disease score, especially with Qyad-7-2 with *p*-value = 0.03 (Figure 3 and Figure 4). For QTLs detected on chromosomes 15 and 18, allele A of the variant AG/GA was associated with a higher AUDPC value while allele G linked with the lower AUDPC value in the population (Figure 5 and Figure 6).

## 4. Discussion

This study selected two parents based on their responses to yam anthracnose disease and created their F_1_-derived recombinant clonal population to assess the functional association of anthracnose resistance and genetic markers using the QTL mapping approach. The recombinant clonal population showed a differential response to the disease-causing organism over the two-year evaluation period. The recombinant population showed quantitative tolerance with a continuous distribution from resistance to the susceptible range with substantial skewness towards resistance. However, no immune or highly resistant clones were identified. Instead, a large number of the clones expressed tolerance reaction to YAD field infestation. The heritability estimate in the present study was high, indicating the proportion of phenotypic variance that is genetic. Similarly, Petro et al. [26] and Bhattacharjee et al. [29] reported high heritability estimates for YAD in water yam.

In an earlier effort, Cormier et al. [32] constructed a high-density genetic map of *D. alata* using 1579 polymorphic SNP markers with a consensus map length of 2613.5 cM. However, our genetic linkage map was built using 3184 SNPs from the GBS platform that spanned a total length of 1460.94 cM representing the most saturated and accurate genetic map for *D. alata* to date. Genetic linkage maps of water yam were also developed using EST-SSRs [29] and AFLPs [5,26]. The genetic linkage map presented in this report will offer a unique opportunity for qualitative and quantitative trait analysis in water yam.

Three studies have been conducted to map QTLs controlling resistance to anthracnose in water yam [5,26,29]. The study by Mignouna et al. [5] and Petro et al. [26] utilized AFLP maps and identified one and nine QTLs, respectively, for anthracnose resistance, explaining 10% and 26–74% of the total phenotypic variation. Bhattacharjee et al. [29] utilized an EST-SSR genetic map for their study and identified a major QTL on linkage group 14, explaining 69% of the total phenotypic variance. Even though the previous studies ordered markers on 20 linkage groups, the absence of a standard genetic map and the different marker systems makes it difficult to compare the location of the detected QTLs in these studies. In the present study, four QTLs located on three chromosomes, accounting for 29.54–39.4% of the total phenotypic variation in the trait, were identified. QTL interaction over the years revealed the stability of three QTL and indicated their potential to predict the specific effect of variation in strain or intensity of strains of *C.*
*gloeosporioides* infestation over the years during the field experimentation.

Furthermore, gene annotation in the QTL regions related to YAD resistance showed known genes that code for plant defense mechanisms. Notably, the region composed of Qyad-7-1 shows the GDSL-like Lipase/Acylhydrolase gene that is reported to regulate systemic resistance to *Alternaria brassicicola* in Arabidopsis [45,46]. Hong et al. [47] also found this gene involved in the defense against drought and *Xanthomonas campestris* pv. Vesicatoria in pepper. Additionally, the protein kinase domain is involved in regulating the resistance against bacterial blight (*Xanthomonas oryzae*) in rice [48] and resistance to the necrotrophic fungal pathogen *Plectosphaerella cucumerina* in Arabidopsis [49] is also present within the QTL region related to YAD in our study. Moreover, the QTL loci on chromosomes 15 and 18 showed the ANTH domain associated with defense against *Pseudomonas syringae* in *Nicotiana benthamiana* and Arabidopsis [50], and the WD domain enhanced the resistance to anthracnose leaf blights in maize caused by *Colletotrichum sublineolum* [51,52]. The F-box protein found within the QTL region of chromosome 15 was reported to be involved in cell death and defense response during the pathogen recognition of *Pseudomonas syringae* and Tobacco mosaic virus in tomato and tobacco [53]. The N-terminal domain within the flanking sequence of the QTL region was involved in the resistance to the downy mildew pathogen *Hyaloperonospora arabidopsidis* in *Arabidopsis* [54]. Therefore, enough evidence exists to confirm that the genes within the flanks of the significant QTLs for anthracnose disease resistance discovered in this study code for response to plant biotic stress.

## 5. Conclusions

This study developed a highly saturated and accurate genetic linkage map for water yam. The linkage mapping approach used in this study identified and mapped QTLs linked to yam anthracnose disease. The QTL regions identified in this study showed six already known genes involved in plant defense. Our results are valuable tools for developing water yam cultivars with quantitative resistance to anthracnose disease. However, these QTLs need to be validated in different genetic backgrounds and environments to be routinely applied in marker-assisted selection in water yam breeding programs in Africa.

## Figures and Tables

**Figure 1 genes-13-00347-f001:**
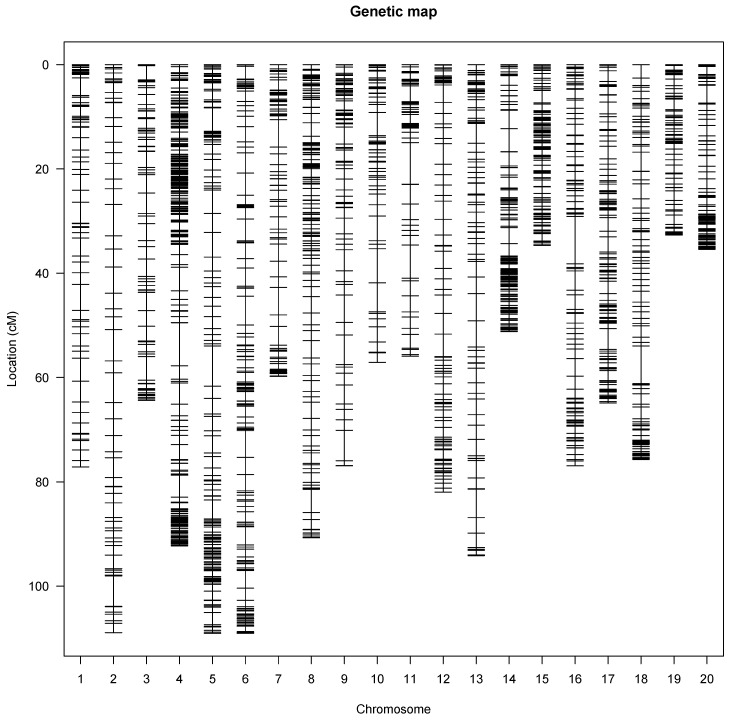
Genetic linkage map showing the SNP distribution across the 20 yam chromosomes. Each vertical line represents a yam chromosome with genetic distance in Kosambi units (cM).

**Figure 2 genes-13-00347-f002:**
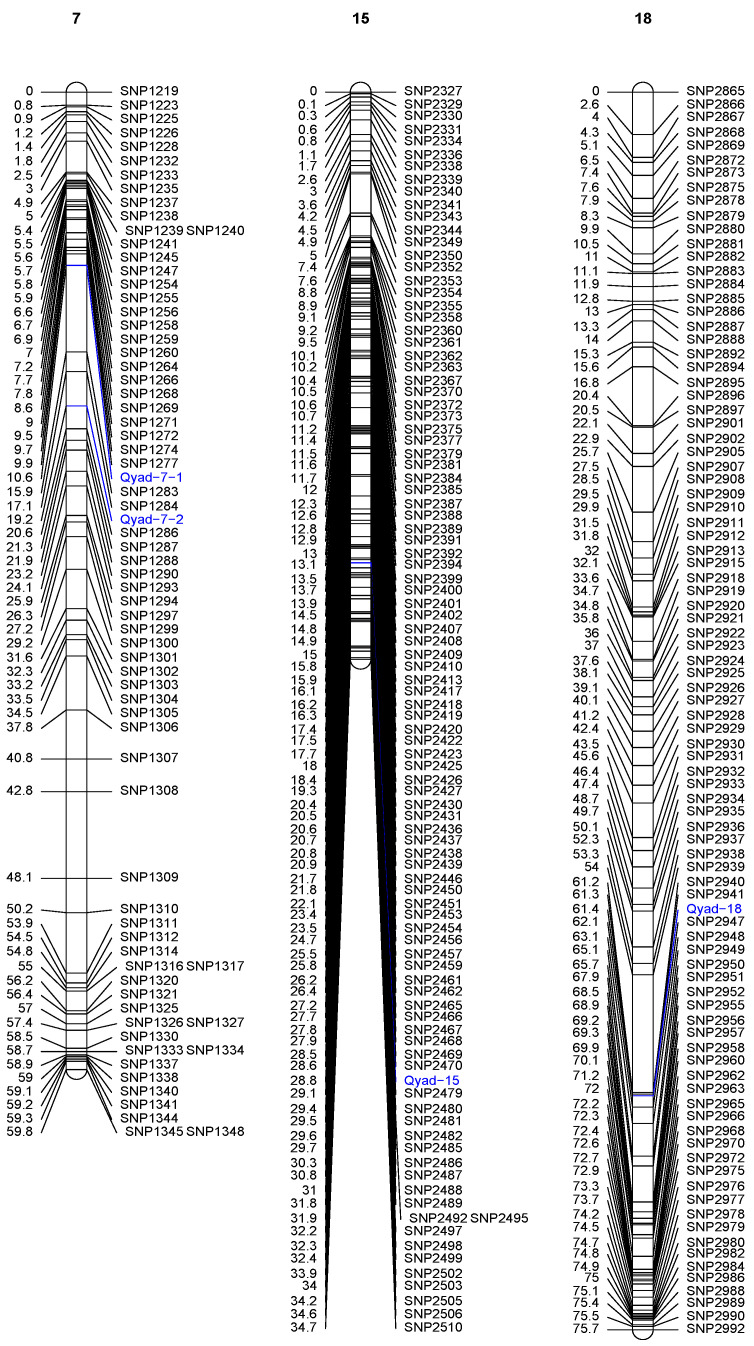
Genetic map of water yam showing significant QTLs associated with yam anthracnose disease resistance. Only those chromosomes where significant QTL are located are displayed. The identified QTLs are highlighted in blue on each chromosome.

**Figure 3 genes-13-00347-f003:**
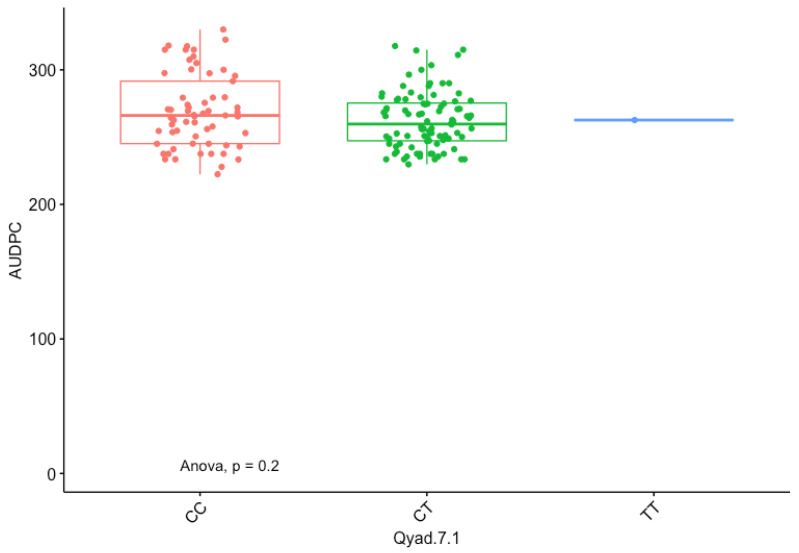
The boxplot showing the effect of the different alleles (variants) of Qyad-7-1 on the AUDPC values. The letters on the X-axis represent alleles (CC, CT, and TT).

**Figure 4 genes-13-00347-f004:**
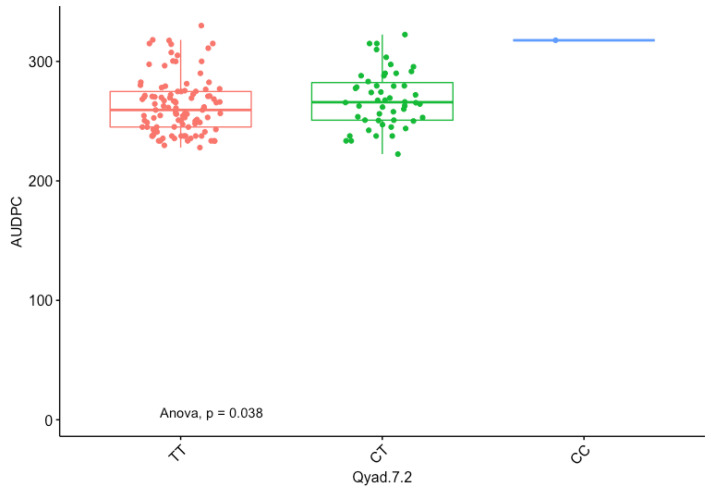
The boxplot displaying the effect of the different alleles (variants) of Qyad-7-2 on the AUDPC estimates. The letters on the X-axis represent alleles (CC, CT, and TT).

**Figure 5 genes-13-00347-f005:**
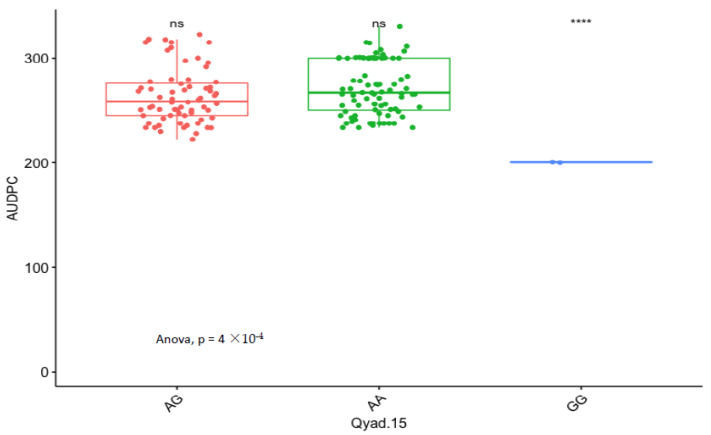
Comparison of the effects of the different alleles (variants) of Qyad-15 on the AUDPC estimates in the study population. The letters on the X-axis represent alleles (AA, AG, and GG), **** statistical significance at *p* values 0.0001 while ns is non-significant.

**Figure 6 genes-13-00347-f006:**
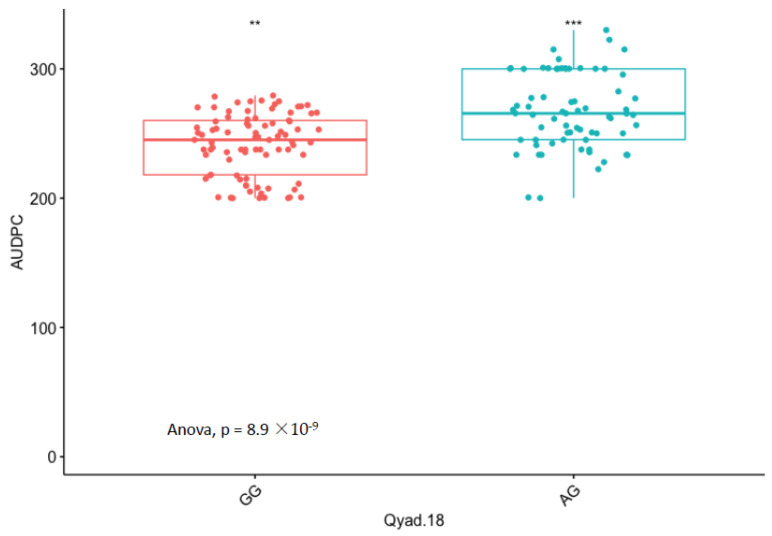
Comparisons of allelic effects of the QTL Qyad-18 on AUDPC estimates in the study population. The letters on the X-axis represent alleles (AG and GG), ** and *** are statistical significance at *p* values 0.05 and 0.001.

**Table 1 genes-13-00347-t001:** Mean squares and heritability estimate for yam anthracnose disease severity in the mapping population.

	Mean Squares			CV	Broad Sense Heritability
Trait	Genotype	Year	Genotype × Year	(%)	(%)
AUDPC	163.01 *	2190.01 **	3371.8 ***	17.6	70.64

AUDPC: area under disease progression curve; *, **, *** significance at 0.05, 0.01, and 0.001 *p*-values, respectively; CV: coefficient of variation.

**Table 2 genes-13-00347-t002:** Distribution of SNP markers and linkage group length (cM) in each chromosome.

Chromosomes	Number of SNPs	Chromosome Length (cM)	Average SNP Distance	Maximum Gap
Chr1	80	80.93	1.95	5.00
Chr2	84	109.17	1.28	6.01
Chr3	115	64.63	0.56	4.48
Chr4	520	92.32	0.16	8.26
Chr5	199	109.19	0.50	7.71
Chr6	191	109.52	0.57	5.54
Chr7	127	59.84	0.48	5.26
Chr8	200	91.39	0.48	4.44
Chr9	124	77.12	0.62	5.80
Chr10	104	57.13	0.55	5.52
Chr11	85	55.95	0.66	7.94
Chr12	125	83.22	0.65	4.30
Chr13	116	95.55	0.78	5.47
Chr14	208	51.447	0.25	4.40
Chr15	180	34.70	0.18	2.39
Chr16	139	77.16	0.61	9.07
Chr17	208	67.44	0.31	3.95
Chr18	129	75.71	0.54	7.23
Chr19	111	33.09	0.28	2.03
Chr20	139	35.48	0.23	3.41
Total	3184	1,460.98		

**Table 3 genes-13-00347-t003:** Summary of significant QTLs detected for yam anthracnose disease resistance in water yam.

Markers	Chr	Pos (cM)	LOD	Add/Dom	CI. Low	CI. High	R^2^ (%)	Putative Genes
Qyad-7-1	7	10.60	4.51	−2.56	10.596	19.217	33.7	DRNTG_08663.1
QTL-7-2	7	19.21	5.28	−5.98	10.596	19.218	29.54	DRNTG_08664.1, DRNTG_23336.1
Qyad-15	15	28.80	4.43	−10.12	10.171	28.817	30.90	DRNTG_14305.1
Qyad-18	18	61.4	4.65	−3.48	61.345	61.432	39.40	DRNTG_18245.1, DRNTG_29617.1

Chr: chromosome; pos: position; LOD: logarithm of odds score; CI: confidence interval; R^2^: % variation explained; Add: additive; Dom: dominance.

**Table 4 genes-13-00347-t004:** Interactions among the detected QTL.

Marker Interactions	df	MS	*p*-Value	Adjusted R-Squared
Qyad-7-1: QTL-7-2	1	55.9	0.835	0.04147
Qyad-7-1: Qyad-15	1	5303.5	0.0456 *	0.02544
Qyad7-1: Qyad-18	1	155.2	0.734	−0.0002131
QTL-7-2: Qyad-15	1	2580.7	0.158	0.04395
Qyad-7-2: Qyad-18	1	6341.0	0.026 *	0.06074
Qyad-15: Qyad-18	1	1408.4	0.309	−0.01079
Qyad-7-1: QTL-7-2: Qyad-15: Qyad-18	3	1247.7	0.068	0.04413

df: degree of freedom; MS: mean square; * statistical significance at *p*-value 0.05.

**Table 5 genes-13-00347-t005:** QTL by environment analysis considering the major QTL.

	Sum Sq	Mean Sq	F Value
Year	174,125	87,062	370.9525
Qyad-7-1	12.25	12.25	0.0002 ***
QTL-7-2	101	101	0.0003 ***
Qyad-15	129	129	0.0001 ***
Qyad-18	2.6	2.6	0.0002 ***
Year × Qyad-7-2	275	112	0.789ns
Year × QTL-7-2	342	78	0.02 *
Year × Qyad-15	278	110	0.226ns
Year × Qyad-18	178	89	0.567ns

MS: mean square; TPVE: total phenotypic variation estimation; ns: non-significant; *, *** statistical significance at *p* values 0.05 and 0.001, respectively.

## Data Availability

Data available at https://yambase.org/breeders/trial/1047 (accessed on 17 April 2018) and https://figshare.com/account/home (accessed on 18 October 2021).

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
