# Peer review of "Identification of QTLs Controlling Resistance to Anthracnose Disease in Water Yam (*Dioscorea alata*)"

_genes, 2022, doi:10.3390/genes13020347_

Round 1

Reviewer 1 Report

The manuscript entitled; “Identification of QTLs Controlling Resistance to Anthracnose Disease in Water Yam (Dioscorea alata)” by Agre et al. describes about the role major novel QTLs to confer anthracnose resistance. The authors have presented his study very well using novel breeding tools and proper statistical analysis. However, I will suggest following changes before its acceptance.

Introduction

Please mention the concrete hypothesis to achieve the objective of current study

Materials and Methods

Authors need to mention how the gene mining was done.

Results

Results and table about gene mining is missing although it was mentioned in abstract and discussion section. I will suggest including supplementary table.

Table 5, if the p-value is less than 0.001 then authors can put three stars (***).

Figure 3-6, please put value for different alleles.

Author Response

Comments and Suggestions for Authors

The manuscript entitled; “Identification of QTLs Controlling Resistance to Anthracnose Disease in Water Yam (Dioscorea alata)” by Agre et al. describes about the role major novel QTLs to confer anthracnose resistance. The authors have presented his study very well using novel breeding tools and proper statistical analysis. However, I will suggest following changes before its acceptance.

Introduction

Query 1

Please mention the concrete hypothesis to achieve the objective of current study

Response

We appreciate suggestion from the reviewer, and we have added hypothesis to achieved the objective presented in this study

Materials and Methods

Query 2

Authors need to mention how the gene mining was done.

Response

Thanks for pointing out this missing information we have provided details on how the genes were searched in the materials and methods

Results

Query 3

Results and table about gene mining is missing although it was mentioned in abstract and discussion section. I will suggest including supplementary table.

Response

We have provided information about the genes in the revised version ad this can be found in the supplementary table S. Also, we have provided methodology used for the gene mining in the materials and methods

Query 4

Response

Table 5, if the p-value is less than 0.001 then authors can put three stars (***).

Response

Thanks for the keen observation we have adjusted this in the revised manuscript

Query 5

Figure 3-6, please put value for different alleles.

Response

We have provided new figures with the p-value as requested for figure 3 -6

Reviewer 2 Report

I have gone through the manuscript. A very informative topic and scientific conclusion have been presented. The findings are significant. Overall, this manuscript is well written. The subject to that: Further recent research on the subject examined should also be included in the relevant section. The materials used in the study were found clearly described in the manuscript. The statistical analyses were also found suitable. Results were found appropriate and given in a logical order. The data is presented in the form of Figures and Tables with satisfactory discussion in the related section.

  1. The introduction should be improved by adding the latest references
  2. Briefly discuss the entire section of materials and methods to be compatible with journal requirements.
  3. Through English Editing

Author Response

I have gone through the manuscript. A very informative topic and scientific conclusion have been presented. The findings are significant. Overall, this manuscript is well written. The subject to that: Further recent research on the subject examined should also be included in the relevant section. The materials used in the study were found clearly described in the manuscript. The statistical analyses were also found suitable. Results were found appropriate and given in a logical order. The data is presented in the form of Figures and Tables with satisfactory discussion in the related section.

Query 1

  1. The introduction should be improved by adding the latest references

Response

We do appreciate suggestion from the reviewer, and we have added recent references in the introduction as well as in the discussion sections

Query 2

  1. Briefly discuss the entire section of materials and methods to be compatible with journal requirements.

Response

Thanks for your suggestion the materials and methods section have been slightly discuss in the discussion section to meet the journal requirements

Query 3

  1. Through English Editing

 Response 3

The revised manuscript has been improved as requested

Round 2

Reviewer 1 Report

I have no more comments